# (S)-10-Hydroxycamptothecin Inhibits Esophageal Squamous Cell Carcinoma Growth In Vitro and In Vivo Via Decreasing Topoisomerase I Enzyme Activity

**DOI:** 10.3390/cancers11121964

**Published:** 2019-12-06

**Authors:** Mengqiu Song, Shuying Yin, Ran Zhao, Kangdong Liu, Joydeb Kumar Kundu, Jung-Hyun Shim, Mee-Hyun Lee, Zigang Dong

**Affiliations:** 1Department of Pathophysiology, School of Basic Medical Sciences, Zhengzhou University, Zhengzhou 450001, China; smq0705@126.com (M.S.); ysyyezi@163.com (S.Y.); zhaorpp1208@126.com (R.Z.); kangdongliu@126.com (K.L.); 2China-US (Henan) Hormel Cancer Institute, No.127, Dongming Road, Jinshui District, Zhengzhou 450008, China; s1004jh@gmail.com; 3The Collaborative Innovation Center of Henan Province for Cancer Chemoprevention, Zhengzhou 450001, China; 4Li Ka Shing Applied Virology Institute, University of Alberta, Edmonton, AB T6G 2R3, Canada; joydeb@ualberta.ca; 5Department of Pharmacy, College of Pharmacy, Mokpo National University, Jeonnam 58554, Korea

**Keywords:** topoisomerase I, (S)-10-Hydroxycamptothecin, esophageal squamous cell carcinoma, patient derived tumor xenograft (PDX)

## Abstract

Topoisomerase (TOP) I plays a major role in the process of supercoiled DNA relaxation, thereby facilitating DNA replication and cell cycle progression. The expression and enzymatic activity of TOP I is positively correlated with tumor progression. Although the anticancer activity of (S)-10-Hydroxycamptothecin (HCPT), a TOP I specific inhibitor, has been reported in various cancers, the effect of HCPT on esophageal cancer is yet to be examined. In this study, we investigate the potential of HCPT to inhibit the growth of ESCC cells in vitro and verify its anti-tumor activity in vivo by using a patient-derived xenograft (PDX) tumor model in mice. Our study revealed the overexpression of TOP I in ESCC cells and treatment with HCPT inhibited TOP I enzymatic activity at 24 h and decreased expression at 48 h and 72 h. HCPT also induced DNA damage by increasing the expression of H2A.X^S139^. HCPT significantly decreased the proliferation and anchorage-independent growth of ESCC cells (KYSE410, KYSE510, KYSE30, and KYSE450). Mechanistically, HCPT inhibited the G2/M phase cell cycle transition, decreased the expression of cyclin B1, and elevated p21 expression. In addition, HCPT stimulated ESCC cells apoptosis, which was associated with elevated expression of cleaved PARP, cleaved caspase-3, cleaved caspase-7, Bax, Bim, and inhibition of Bcl-2 expression. HCPT dramatically suppressed PDX tumor growth and decreased the expression of Ki-67 and TOP I and increased the level of cleaved caspase-3 and H2A.X^S139^ expression. Taken together, our data suggested that HCPT inhibited ESCC growth, arrested cell cycle progression, and induced apoptosis both in vitro and in vivo via decreasing the expression and activity of TOP I enzyme.

## 1. Introduction

Esophageal cancer (EC) is the eighth most common cancer globally and ranks sixth among the worst-prognosis cancers [1]. Clinically, EC is quite aggressive in nature and has a poor survival rate [1]. Esophageal squamous cell carcinoma (ESCC), the major subtype of EC, has a 79% incidence in Asia [2]. Although the survival time of EC patients has improved in the past 30 years, it still remains poor compared to the other cancer types [3]. Because of apparently late diagnosis and the lack of a well-defined drug target, EC patients poorly respond to conventional chemoradiotherapy or chemoradiotherapy plus surgery. Thus, the identification of a valid drug target and the design of molecular target-based anticancer therapy would be a rational strategy to limit mortality from EC.

Topoisomerase I (TOP I) is an essential enzyme in both prokaryotes and eukaryotes [4]. TOP I is involved in the process of supercoiled DNA relaxation and the alleviation of the DNA helical constraints [5]. TOP I binds to the supercoiled DNA and subsequently, the cleaved one strand of the duplex DNA, then, a nick will be created at this site to allow the DNA to untwist and relax [6]. The participation of TOP I facilitates the process of DNA replication and the formation of RNA:DNA hybrids. The lack of TOP I or functional inactivation of the enzyme leads to DNA double-strand break (DSB) formation [7]. Elevated expression and enzymatic activity of TOP I have been reported to be closely related with cancer development, and it is a therapeutic target for the camptothecin (CPT) family [8]. The aberrantly high level of TOP I expression in paraffin-embedded tissues sections of different primary tumors including ovarian carcinoma and gastric cancer have been reported [9,10,11,12,13]. Thus, TOP I enzyme may be an ideal indicator of tumor progression and a potential drug for CPT family chemotherapeutic agents [14].

(S)-10-Hydroxycamptothecin (HCPT) (The chemical structure was indicated in Appendix A), which is isolated from a Chinese tree *Camptotheca cuminata*, has been developed as a TOP I specific inhibitor [15]. HCPT has been reported to have broad anti-cancer activity in murine leukemia cells [16], human neuroblastoma [17], and colon cancer [18]. The nanosuspensions or polymorphic nanoparticle formulations of HCPT are the two new formulations that helped to increase the stability of the original HCPT and enhanced its anti-tumor efficacy [19,20]. Based on its broad anti-cancer spectrum, the evaluation of the effect of HCPT on ESCC cell proliferation and the capability of developing this compound for clinical management of EC would be a rational approach. Thus, the present study was designed to examine the effect of HCPT on the growth of ESCC cells’ growth in culture as well as in vivo using a patient-derived xenograft (PDX) tumor model in mice. Here, we report that HCPT blunted the enzymatic activity of TOP I and the treatment with HCPT attenuated proliferation, arrested the cell cycle and induced apoptosis in ESCC cells, and diminished the growth of ESCC PDX models in vivo. Taken together, our study reveals the potential of HCPT as an anticancer therapeutic agent for ESCC.

## 2. Results

### 2.1. TOP I Enzyme Acts as an Indicator of ESCC

Since TOP I is a potential biomarker of tumor progression in many cancers, we performed immunohistochemical (IHC) analysis of TOP I expression in ESCC tissues. The expression of TOP I was significantly elevated in ESCC tumor tissues as well as in tumor-adjacent tissues as compared to normal tissues (Figure 1A,B). The transcripts of TOP1 also dramatically increased in esophageal cancer patients’ tissues compared to normal tissues (Appendix A) (Data obtained from http://gepia.cancer-pku.cn/). Data downloaded from the TCGA database showed that the TOP I gene is overexpressed in different stages of ESCC progression in patients as compared to its expression in normal tissues (Figure 1C). Moreover, patients with a high expression of TOP I had a relatively shorter overall survival time than those with low expression of the gene (*p* = 0.014) (Figure 1D) (Data obtained from http://gepia.cancer-pku.cn/). Western blot was also performed to identify the expression of TOP I in cultured ESCC cells. The TOP I was highly expressed in most of the ESCC cell lines, especially in KYSE410, KYSE510, KYSE30, and KYSE450 cells, however its level was relatively low in normal esophageal epithelial cell SHEE (Figure 1E, Appendix A).

### 2.2. HCPT Inhibits the Proliferation of Esophageal Squamous Cell Carcinoma Cells

In order to examine the effects of HCPT on ESCC cells, we selected four kinds of ESCC cell lines (KYSE410, KYSE510, KYSE30, and KYSE450), which contained higher levels of TOP I protein for cell proliferation assay (Figure 1E). The data indicated that HCPT treatment significantly decreased the proliferation of ESCC cells in a time- and concentration-dependent manner. The effective concentration (EC_50_) of HCPT ranged between 40 nM and 320 nM (Figure 2A). However, HCPT did not cause any cytotoxicity on normal esophageal epithelial cell SHEE (Appendix A). Moreover, HCPT dramatically inhibited the foci formation at a concentration of 40 nM, which also showed significant inhibition of cell proliferation (Figure 2B,C). In the anchorage-independent cell growth assay, HCPT showed a strong inhibitory effect on colony formation consistent with MTT and foci assay in these ESCC cell lines (Figure 2D,E).

### 2.3. HCPT Interrupts G2/M Cell Cycle Transition and Induces Apoptosis in ESCC Cells

The incubation of KYSE410, KYSE510, KYSE30, and KYSE450 cells with HCPT led to cell cycle arrest at the G2/M phase (Figure 3A), inhibition of cyclin B1 expression, and elevation of p21 expression (Figure 3B, Appendix A). Treatment with HCPT caused ESCC cells to undergo apoptosis as revealed by annexinV/PI staining. The total apoptosis cell number was collected based on early apoptosis (annexinV+/PI− gate) and late apoptosis (annexinV+/PI+ gate). HCPT significantly induced cell apoptosis compared to the DMSO treatment control, and this phenomenon happened both at early and late apoptosis (Figure 3C, Appendix A). Subsequent Western blot analysis showed changes in apoptosis markers in four candidate cell lines after HCPT treatment. The expression of cleaved PARP, cleaved caspase-3, cleaved caspase-7, Bax, and Bim, which are markers of apoptosis, was significantly increased after HCPT treatment for 72 h, while that of the anti-apoptotic protein Bcl-2 was markedly decreased (Figure 3D, Appendix A).

### 2.4. HCPT Decreases TOP I Enzyme Activity and Inhibits its Expression in ESCC

We next assessed the effect of HCPT on the TOP I enzyme activity in ESCC. DNA electrophoresis showed that the amount of supercoiled DNA was obviously increased as compared to the control group in all of the four ESCC cell lines after treatment of HCPT for 24 h (Figure 4A). The data indicated that HCPT treatment significantly inhibited the activity of TOP I enzyme and interrupted the re-ligation of DNA strand. Western blot analysis of lysates from ESCC cells treated with HCPT for 48 h and 72 h showed decreased TOP I expression in all of the four candidate cell lines (Figure 4B, Appendix A). We also found that the expression of rH2A.X was elevated 48 h post HCPT treatment as compared with the DMSO-treated group, indicating that DNA damage occurred because of the lack of sufficient TOP I activity (Figure 4C, Appendix A).

### 2.5. HCPT Attenuated ESCC PDX Tumor Growth in Mice

PDX tumors in mice are an experimental model for pre-clinical study to assess the potential of anti-tumor activity of putative drug molecules [21]. To examine the possible inhibitory effects of HCPT on PDX tumor growth, patient-derived LEG104 and LEG110 ESCC tumors with high expression levels of TOP I were adopted in this investigation. The treatment of PDX tumors grown in mice with HCPT (4 mg/kg and 8 mg/kg body weight) decreased the tumor volume and tumor weight significantly (Figure 5A–D). The weight and size of tumors were also dramatically decreased after HCPT administration for 18 or 29 days (Figure 5B,C,E,F). While eliciting antitumor activity in PDX tumor-bearing mice, HCPT application did not exhibit any remarkable signs of toxicity in mice (Appendix A). The IHC and HE staining of the PDX tumor sections showed more regression of tumor cells and changes in morphology, such as cell enlargement, caryolysis, variable nucleus size, nuclear pyknosis, and nucleolar margin blurring in the treatment group in comparison with the vehicle group (Figure 5G). Darker staining in the cytoplasm by eosin in the HCPT-treated groups indicated apoptosis phenomenon which was not evident in the vehicle group (Figure 5G). Moreover, there was a significant decrease in the expression of Ki-67 and elevation in cleaved caspase-3, which indicated that the cell proliferation was inhibited and apoptosis was induced after HCPT application. The IHC analysis of HCPT-treated tumors also showed decreased TOP I expression and increased phosphorylated rH2A.X level (Figure 5G,H, Appendix A).

## 3. Discussion

The aberrant elevated expression of DNA TOP I at both the protein and mRNA levels has been frequently observed in several human malignancies compared to normal tissues. Therefore, making TOP I a potential target for anticancer medicines is important [14]. TOP I protein level has been reported to be highly expressed in various ESCC cell lines and the protein expression index demonstrated a significant correlation with the IC_50_ value of camptothecin-11 treatment, which did not show any correlation with the mRNA level of TOP I [22]. We examined the effects of 10-HCPT, a derivative of camptothecin, on the growth of ESCC cells in culture cells as well as in vivo and elucidated its underlying mechanism. In our findings, IHC results of ESCC tissue array indicated that TOP I protein was overexpressed in tumors and tumor-adjacent tissues compared with normal esophageal epithelial tissues (Figure 1A,B). This finding was in good agreement with the elevated expression of TOP I in ESCC cells in culture (Figure 1E, Appendix A). Meanwhile, the GEPIA database indicated that the transcripts of TOP1 were significantly increased in ESCA patient samples (Appendix A). The positive correlation between TOP I expression and the ESCC cancer stage, especially stage IV (Figure 1C), make this gene a potential target for developing therapy for ESCC. Patient survival time also exhibited significant difference between the TOP I-high and TOP I-low expression groups, demonstrating that high TOP I expression leads to poor outcomes.

TOP I was reported to be a molecular target of chemotherapeutic medicine of the camptothecin (CPT) family [8]. Compared to CPT, HCPT showed more profound anti-cancer effects with relatively less toxicity. According to several other studies, HCPT exhibited a broad spectrum of anti-cancer effect in a variety of solid tumors in vitro and in vivo [23,24,25]. There are also other camptothecin analogues that were used to inhibit esophageal squamous cell carcinoma growth, like gimatecan and irinotecan. Compared with the first-line clinical agent, irinotecan, gimatecan showed stronger anticancer activity. Both gimatecan and irinotecan functioned through topoisomerase inhibition and subsequent induction of DNA damage, S-phase cell cycle arrest, and induction of apoptosis [26]. In a phase III clinical trial, the progression-free survival (PFS) (*p* = 0.006) and objective response rate (ORR) (*p* = 0.002) of recurrent or metastatic ESCC were prolonged in the irinotecan plus intravenous infusion of irinotecan (S-1) [160 mg/m^2^] group compared to the S-1 monotherapy group, leading to the possibility of TOP I inhibitor on ESCC clinical treatment [27]. HCPT was reported to decrease cell proliferation and induce apoptosis or autophagy in colon cancer cells [18] and non-small cell lung cancer (NSCLC) cells [28]. In agreement with these reports, we found that HCPT suppressed ESCC cells proliferation dramatically in a time- and concentration-dependent manner in both the MTT assay and foci assay (Figure 2A–C). The anti-tumor effect of HCPT was further supported by decreased size and number of colonies in anchorage-independent cell growth assay (Figure 2D,E). It was reported in earlier studies that HCPT induced S phase cytotoxicity via inducing replication fork collision and finally leading to G2/M phase cell cycle arrest [15,29]. We confirmed the finding through flow cytometry analysis of the cell cycle after HCPT treatment for 24 h. HCPT application destroyed S phase cell cycle progression and inhibited the G2/M phase of cell cycle transition in ESCC cell lines, attenuated cyclin B1 protein expression, as well as increased the expression of P21 protein (Figure 3A,B, Appendix A). Our flow cytometry analysis results of the induction of apoptosis in ESCC cells upon treatment with HCPT (Figure 3C, Appendix A) was also well correlated with previous reports of apoptosis induction by HCPT in SMS-KCNR human neuroblastoma cells [17], human osteosarcoma cells [30], colon cancer cells [18], and hepatoma Hep G2 cells [31]. The elevated expression of cleaved PARP, cleaved caspase-7, cleaved caspase-3, and Bax and the decreased expression of Bcl-2 were the molecular probes to the apoptosis induction by HCPT. Moreover, the decreased expression of long form of Bim or the increased expression of short form of Bim also indicated apoptosis post HCPT treatment (Figure 3D, Appendix A).

Accumulating data indicated that HCPT formed a complex with DNA and TOP I, resulting in double strand DNA breakage and cell death directly [15,32]. Since human TOP I is essential for topological stress releasing and topology modulation, the damage of TOP I function will lead to supercoiled DNA because the DNA strands cannot release superhelix during transcription and replication [4]. Thus, the amount of supercoiled DNA will reflect the enzyme activity of TOP I in double strand DNA superhelix locally unwinding. Treatment of cells for 24 h with HCPT as a specific inhibitor of TOP I showed obvious supercoiled DNA formation compared to the DMSO control group in ESCC cells (Figure 4A). This finding demonstrated that the function of human TOP I activity was diminished after HCPT treatment. Besides decreasing the TOP I activity, HCPT treatment also reduced the protein expression of TOP I in ESCC cells as revealed by immunoblot analysis (Figure 4B, Appendix A). This decrease in TOP I protein expression was more obvious at 72 h in ESCC cells treated with HCPT (Figure 4B, Appendix A).

Histone H2A.X is a variant histone that is responsible for DNA-damage repair [33]. Its phosphorylation at Ser139 is required for the recognition of double-stranded breaks (DSBs) [34]. Histone H2A.X^S139^ was indicated to be characteristic of higher genome instability and its increase in protein expression indicates DNA double-strand breaks and DNA repair emergency [35]. Due to the important function of TOP I in DNA and chromatin conformations, the application of the camptothecin family inhibitor finally induced cellular DNA damage and cell death [36]. Based on this knowledge, we assessed any possible changes in H2A.X^S139^ expression following the administration of HCPT. The exposure of ESCC cells to HCPT for 48 h increased the H2A.X^S139^ level, which indicated the enhancement of DNA damage after the inhibitor application (Figure 4C, Appendix A).

The patient-derived xenograft (PDX) tumors reflect the gene expression profiles and epigenomes of the original tumors [37], making the xenograft an ideal model for anti-cancer medicine screening. In our study, ESCC PDX LEG104 and LEG110 were used to evaluate the chemotherapeutic ability of HCPT. The administration of HCPT inhibited ESCC PDX tumor growth in both of ESCC cases, and the tumor weight was dramatically decreased (Figure 5A–F). However, HCPT treatment did not show remarkable changes in mouse body weight and pathological changes in the liver and spleen (Appendix A). Immunohistochemical analysis of PDX xenograft tumor tissues from HCPT- or vehicle-treated groups also showed reduced expression of Ki-67 and TOP I and increased expression of cleaved caspase-3 and H2A.X^S139^ in HCPT-treated tumors (Figure 5G,H, Appendix A), further supporting the anti-proliferative and apoptosis inducing effect of HCPT on ESCC. Of note, the animal experiments in our study are still limited for statistical assessment, however it indicated the way for ESCC clinical treatment. The activity of TOP I enzyme was inhibited as an early stage phenomenon after HCPT administration and the DNA damage, cell cycle arrest, and apoptosis induction were subsequently triggered as a final consequence in regressing the size of PDX tumors.

In conclusion, TOP I is highly expressed in human ESCC tumors and tumor-adjacent tissues and treatment with HCPT attenuates cell proliferation, inhibits G2/M phase cell cycle transition, and induces apoptosis in ESCC cells in culture via inhibition of the expression and activity of TOP I enzyme (Figure 6). Moreover, HCPT suppresses ESCC PDX tumor growth in vivo, which is associated with reduced expression of TOP I and other cell proliferation markers and elevated expression of apoptosis markers. Thus, TOP I appears as a valid anticancer drug target and HCPT, as an inhibitor of TOP I, may be considered for further clinical development for ESCC therapy.

## 4. Materials and Methods

### 4.1. Reagents

HCPT (purity > 98%) was purchased from Meilun Biotechnology Co., Ltd. (CAS: 19685-09-7, Dalian, China) and was analyzed and authenticated by high-performance liquid chromatography. Thiazolyl blue tetrazolium bromide (MTT) powder and Crystal violet powder were purchased from Solarbio Technology Co., Ltd. (Beijing, China). TOP I antibody was purchased from Abcam and antibodies to detect apoptosis such as: cleaved PARP (Cat: 9664), cleaved caspase-3 (Cat: 5625), cleaved caspase-7 (Cat: 8438), Bim (Cat: 2933), Bax (Cat: 5023), and Bcl-2 (Cat: 15071) as well as p21 (Cat: 2947) were purchased from Cell Signaling Technology (Beverly, MA, USA).

### 4.2. Cell Culture

KYSE410, KYSE510, KYSE30, and KYSE450 human esophageal cancer cell lines were purchased from the Type Culture Collection of the Chinese Academy of Sciences (Shanghai, China). Normal human immortalized esophageal epithelial cell SHEE was donated by Dr. Enmin Li in the Laboratory of Tumor Pathology (Shantou University Medical College, Shantou, Guangdong, China) [38]. The cells were cultured in RPMI-1640 contained with streptomycin (100 μg/mL), penicillin (100 units/mL), and 10% FBS (BI, Kibbutz, Israel) in an incubator with 5% CO_2_ and 37 °C atmosphere. SHEE (N1217) was donated by Dr. Enmin Li (Laboratory of Tumor Pathology, Shantou University Medical College, Shantou, Guangdong, China). All of the cells had passed the STR profiling and had been identified as correct cell lines. Each vial of frozen cells was thawed and maintained in culture within 10 passages.

### 4.3. MTT Assay

Cells were seeded in 96-well plates for MTT assay according to cell growth characteristics (3 × 10^3^ cells per well for KYSE410, 1.5 × 10^3^ cells per well for KYSE510, 2 × 10^3^ cells per well for KYSE30, 1.5 × 10^3^ cells per well for KYSE450). Cells were treated with different concentrations of HCPT or vehicle. After incubation for 24, 48, and 72 h, cell proliferation was measured by MTT (0.5 mg/mL) reagent, separately.

### 4.4. Foci Formation Assay

KYSE410, KYSE510, KYSE30, and KYSE450 cells were used to perform Foci formation assay following the protocols described below. Cells (1 × 10^3^ cells/well) were seeded into 6-well plates and incubated for 24 h. The cells were then treated with different concentrations of HCPT or vehicle for another 7 days. The cell culture media was changed every three days. Foci were washed by PBS and then fixed and stained with 0.5% crystal violet solution.

### 4.5. Anchorage-Independent Cell Growth

Agar mix (0.5%) was prepared to form a base layer, which contained vehicle and HCPT (40, 80, and 160 nM) in the 6-well plate. Then, cells (8 × 10^3^ cells/well) suspended in complete medium were added to 0.3% agar with vehicle, 40, 80, and 160 nM HCPT in a top layer over the base layer. Each of the agar plates were cultured and maintained in the incubator for 3 weeks and pictures were taken at the third week by microscope. Then, colonies were counted and summarized using the Image-Pro Plus software (v.6.0) program (Media Cybernetics, Silver Spring, MD, USA).

### 4.6. Cell Cycle Analysis

Cells (3.5 × 10^5^ cells per dish for KYSE410, 2 × 10^5^ cells per dish for KYSE510, KYSE30, KYSE450) were seeded into 6-well plates and incubated for 24 h. Then, media was replaced with FBS- and-antibiotic-free fresh RPMI-1640 media for 12 h. Different concentrations of HCPT or vehicle were added to the cells and incubated for 48 h, then the cells were harvested together with supernatant by trypsin digestion. The cells were fixed with 70% ethanol at −20 °C overnight and washed by pre-cold PBS and then re-suspended with 250 μL 0.6% Triton X-100 solution. Cells were then incubated with RNase A (200 μg/mL) for 1 h at room temperature followed by cell cycle analysis using a BD FACSCalibur Flow Cytometer (BD Biosciences, San Jose, CA, USA) after propidium iodide (PI, 20 μg/mL) staining for 15 min at 4°C.

### 4.7. Annexin V Apoptosis Assay

Cells (2 × 10^5^ cells per dish for KYSE410, 1 × 10^5^ cells per dish for KYSE510, KYSE30, and KYSE450) were seeded into 6-well plates. After incubation for 24 h, cells were treated with HCPT (0, 80, 160, or 320 nM) or vehicle for 72 h and harvested together with cell culture media for apoptosis detection. Harvested cells were then incubated with Annexin V binding buffer for 10 min at room temperature followed by PI staining in dark atmosphere. Apoptosis was detected by a BD FACSCalibur Flow Cytometer (BD Biosciences, San Jose, CA, USA).

### 4.8. Western Blotting

Protein from KYSE410, KYSE510, KYSE30, and KYSE450 cells treated with or without HCPT were extracted using NP-40 cell lysis buffer (50 mM Tris pH 8.0, 150 mM NaCl, 0.5–1% NP-40, Protease inhibitor cocktail, dephosphorylate inhibitor tablets, and 1 mM phenylmethylsulfonyl fluoride (PMSF)). Bichinconinic acid (BCA) assay kit (Solarbio, Beijing, China) was used for protein quantification and the same amount of protein were prepared for each sample according to the protein concentration. Sodium dodecyl sulphate-polyacrylamide gel electrophoresis (SDS-PAGE) was performed to separate different proteins and then the proteins were transferred onto polyvinylidene fluoride membranes, followed by blocking with 5% non-fat milk for 1 h at room temperature. Subsequently, appropriate primary antibodies were incubated at 4°C overnight, followed by incubation with appropriate HRP-linked secondary antibodies for 2 h at room temperature. Blots were visualized by the enhanced chemiluminescence (ECL) detection reagent.

### 4.9. Immunohistochemical (IHC) Analysis

Paraffin-embedded sections of human esophageal squamous cell carcinoma tissue microarray and PDX tumors from mice were subjected to IHC analysis. About 5 μm of thickness of the sections were prepared for each tumor sample. The specimens were blocked with 5% goat serums following antigen unmasking by sodium citrate solution and were first incubated with appropriate primary antibodies at 4°C overnight, followed by secondary antibodies for 2 h at room temperature. The sections were then incubated with 3,3’-diaminobenzidine (DAB) solution to visualize the staining of the target proteins by counterstaining with hematoxylin, dehydrating through a graded series of alcohol into xylene, and being mounted under glass coverslips.

### 4.10. TOP I Assay

This experiment was performed following the protocol of TOP I assay kit (TopoGEN, Florida, USA). Cells treated with HCPT or vehicle for 48 h were harvested together with media and dispersed cell pellets by TEMP buffer (10 mM Tris-HCL, pH = 7.5, 1 mM EDTA, 4 mM MgCl_2_, 0.5 mM PMSF). Nuclear pellets were re-suspended with TEP buffer (same as TEMP but lacking MgCl_2_) following homogenizing and centrifuging. Then, equal volume of 1 M NaCl was added to each tube to get the supernatant by ultracentrifuge. Then, the TOP I activity assay was performed using all of the reactants containing DNA, test extract, Tris-Glycine-SDS (TGS) buffer, and H_2_O to reach a final volume of 20 μL. The mixture was incubated at 37 °C for 30 min and the reaction was terminated by adding 5× stop buffer. The samples were directly loaded onto a 1% agarose gel for electrophoresis until the dye front of bromophenol blue was about 70% down the gel. The gel was then stained with Ethidium bromide (EB) solution for 15 min and then performed photo-documentation immediately.

### 4.11. PDX Tumor Model in Mice

Six week old female severe combined immunodeficient (SCID) mice (Vital River Labs., Beijing, China) were adopted in this study. Tumor tissues from patients were implanted subcutaneously on the mice and the mice were divided into 3 groups (*n* = 10/group) when the tumor mass reached an average volume of 100 mm^3^. The mice in different groups were treated with vehicle or HCPT (4 mg/kg or 8 mg/kg), respectively, twice per week by paraneoplastic injection. The injectable suspension of HCPT was made by mixing HCPT with 10% DMSO, 40% PEG-400, and 50% PBS. The tumor volume of each mice was measured twice per week and then calculated by the following formula: tumor volume (mm^3^) = length × width × height × 0.52. Mice were euthanized and tumors were extracted until the average tumor volume reached 1000 mm^3^. This study was approved by the Ethics Committee of Zhengzhou University (Zhengzhou, Henan, China) (ZZUHCI-2019012) and the patients whose tumor samples were used in this study were completely informed and provided consent.

### 4.12. Statistical Analysis

Data in this paper were shown as mean ± standard deviation (SD). Student’s *t*-test or one-way ANOVA was performed for *p* value analysis, where <0.05 was considered to be statistically significant. All of the statistical analyses were shown as * *p* < 0.05, ** *p* < 0.01, and *** *p* < 0.001 compared to the control group, respectively.

## 5. Conclusions

Our study identified that TOP I acts as an indicator in ESCC, TOP I enzyme activity suppression by HCPT attenuated cell proliferation, inhibited G2/M phase cell cycle transition, and induced apoptosis in ESCC cells in vitro and decreased ESCC PDX tumor growth in vivo. These findings mean that TOP I appears to be a valid therapeutic target and HCPT may be considered for further clinical application in ESCC.

## Figures and Tables

**Figure 1 cancers-11-01964-f001:**
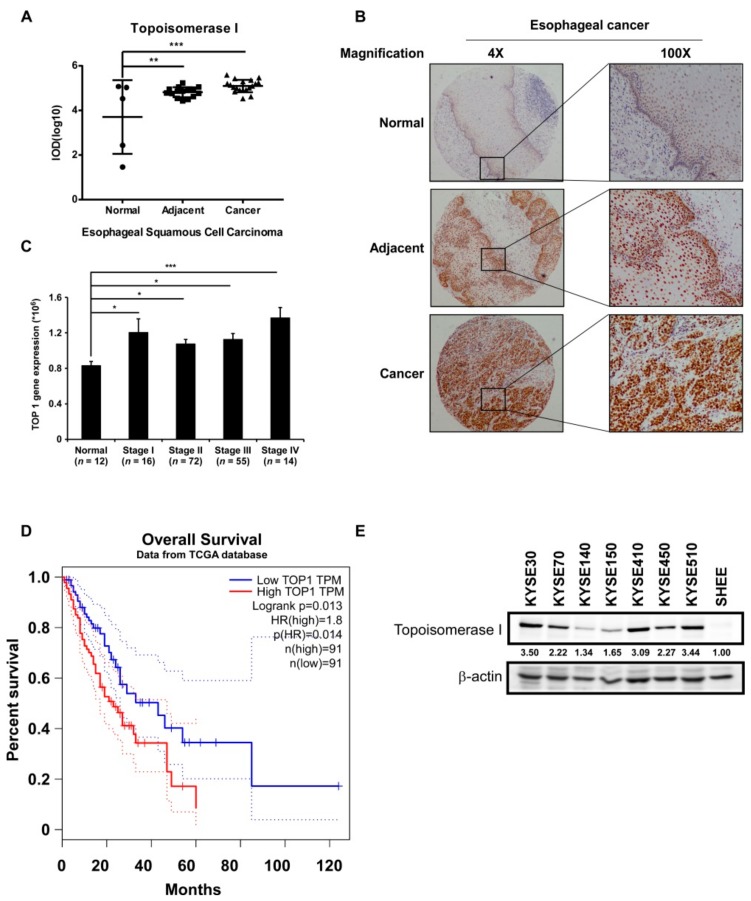
TOP I enzyme acts as an indicator of esophageal squamous cell carcinoma (ESCC). (**A**) Quantitation results of Topoisomerase (TOP) I immunohistochemical (IHC) staining on ESCC tissue array. Data was shown in the value of log10 (IOD). **, *p* < 0.01; ***, *p* < 0.001 compared to normal tissues. (**B**) Images of IHC staining on esophageal normal (5 cases), adjacent (15 cases), and cancer (19 cases) tissues, separately (40× and 100× magnification). (**C**) TOP1 gene expression analysis in esophageal normal tissues and different stage cancer tissues (Data downloaded from TCGA database). *, *p* < 0.05; ***, *p* < 0.001 compared to normal tissues. (**D**) Overall survival time of patients with high or low expression of TOP I gene (data obtained from http://gepia.cancer-pku.cn/). (**E**) The expression of TOP I in different kinds of ESCC cell lines was evaluated by Western blot assay. β-actin was used as an internal reference control.

**Figure 2 cancers-11-01964-f002:**
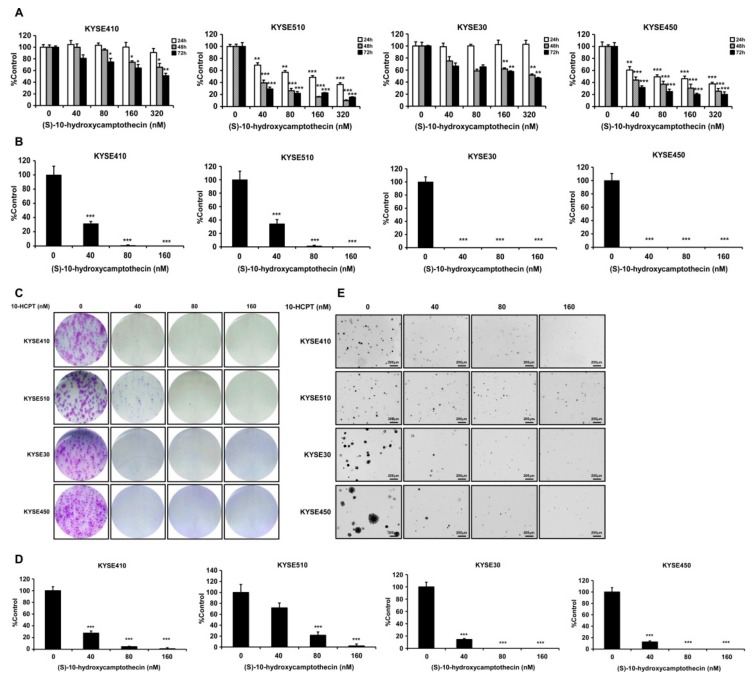
HCPT inhibits esophageal squamous cell carcinoma cells’ proliferation. (**A**) Cells’ proliferation of KYSE410, KYSE510, KYSE30, and KYSE450 post HCPT (0, 40, 80, 160, and 320 nM) treatment were detected by MTT assay. Data were shown compared with the dimethyl Sulfoxide (DMSO) treated group. *, *p* < 0.05; **, *p* < 0.01; ***, *p* < 0.001 compared to the controls. (**B**) Foci formation of ESCC cells were performed in 6-well plates with HCPT (0, 40, 80, and 160 nM) application for 7 days. The colonies number was analyzed and summarized, and the data were shown compared with the DMSO treated group. ***, *p* < 0.001 compared to controls. (**C**) Images of crystal violet stained foci after HCPT (0, 40, 80, and 160 nM) treatment for 7 days. (**D**) Anchorage-independent cell growth assay was performed to evaluate the effect of HCPT (0, 40, 80, and 160 nM) on cell growth. Colonies were captured and the number was counted after 3 weeks; the results are presented as treated group compared with the control group. ***, *p* < 0.001. (**E**) Representative pictures of colonies after HCPT treatment on KYSE410, KYSE510, KYSE30, and KYSE450 cells. Three independent repeats were performed for each experiment and were statistically analyzed.

**Figure 3 cancers-11-01964-f003:**
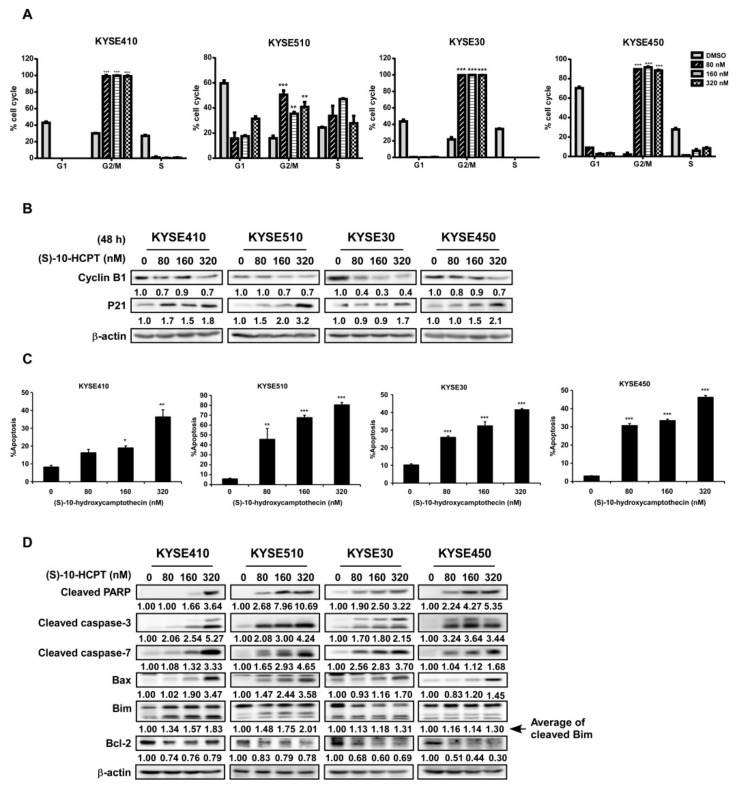
HCPT interrupts ESCC cells’ G2/M cell cycle transition and induces apoptosis. (**A**) Cell cycle analysis by flow cytometry on ESCC cells after HCPT (0, 80, 160, and 320 nM) treatment for 12 h. Statistics of cell cycle distribution were shown as % cell cycle. **, *p* < 0.01; ***, *p* < 0.001 compared to DMSO controls. (**B**) Western blotting analysis to detect the expression of G2/M phase markers cyclin B1 and p21 after HCPT (0, 80, 160, and 320 nM) treatment in ESCC cells. β-actin was used as an internal reference control. (**C**) Cell apoptosis were checked by flow cytometry using annexin V and the PI double staining method. The results were summarized and shown as % apoptosis. *, *p* < 0.05; **, *p* < 0.01; ***, *p* < 0.001 compared to the controls. (**D**) Cells were treated with HCPT (0, 80, 160, and 320 nM) for 72 h. Lysates were harvested and analyzed by Western blotting for the expression of apoptosis markers as indicated. β-actin was used as an internal reference control. Three independent repeats were performed for flow cytometry detection and were statistically analyzed.

**Figure 4 cancers-11-01964-f004:**
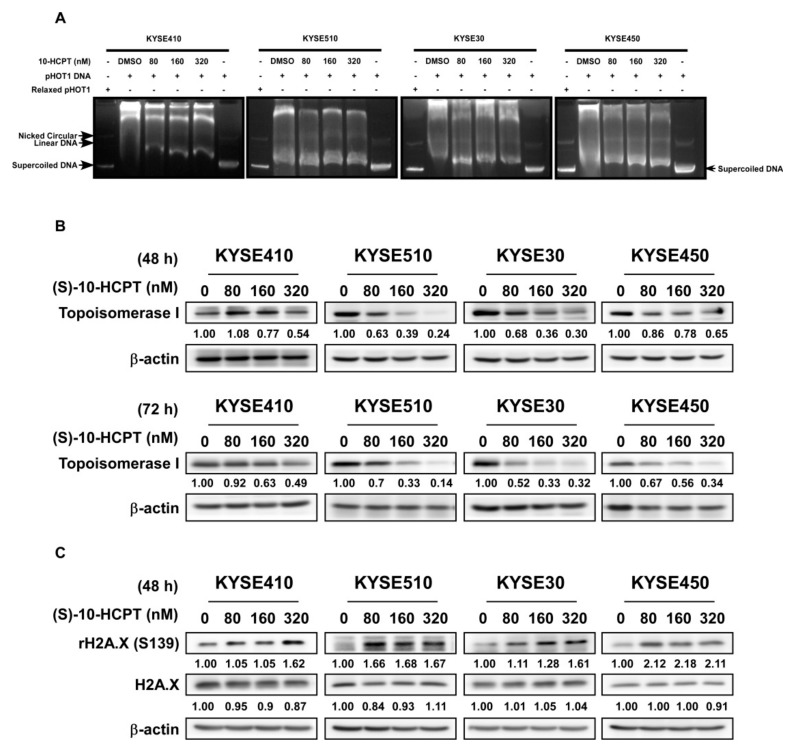
HCPT decreases TOP I enzyme activity and inhibits its expression in ESCC. (**A**) Cell lysates were analyzed after HCPT treatment with the indicated concentration for 24 h. TOP I enzyme was extracted following the producer’s protocol (www.topogen.com). The activity of TOP I was measured and visualized by DNA electrophoresis as the indicated amount of supercoiled DNA. (**B**) ESCC cells were harvested and lysated with HCPT treatment for 48 and 72 h and the expression of TOP I was visualized by Western blot method. β-actin was used as an internal reference control. (**C**) Phosphorylation of rH2A.X at serine 139 site indicated DNA foci formation and damage, which is the late-stage phenomenon after TOP I activity was broken down. Western blot was carried out for developing the expression of H2A.XS139 after HCPT treatment at 48 h in the indicated concentration. β-actin was used as an internal reference control. Three independent repeats were performed for each experiment.

**Figure 5 cancers-11-01964-f005:**
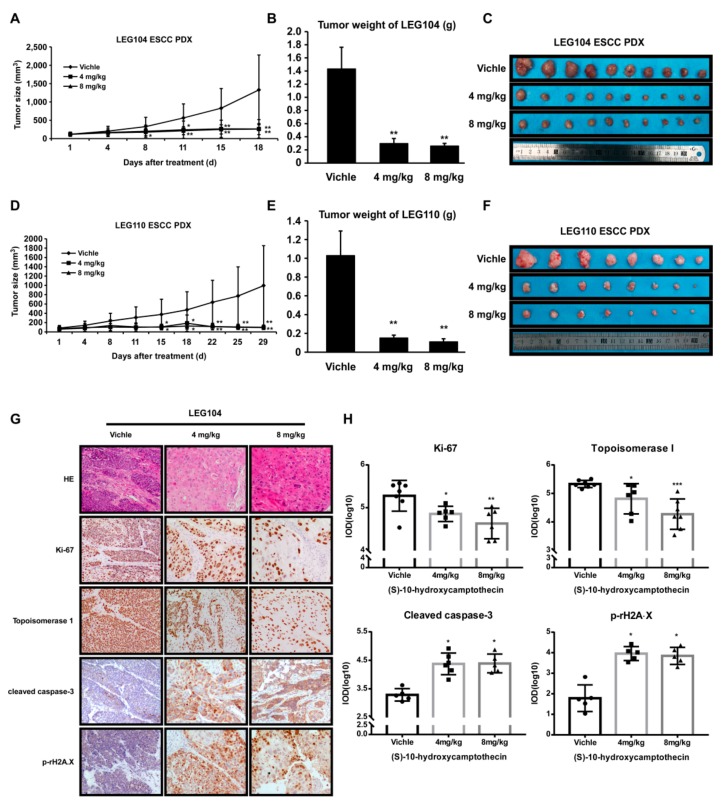
HCPT attenuated ESCC patient-derived xenograft growth in vivo. The effect of (S)-10-Hydroxycamptothecin on the growth of ESCC PDX was plotted over 18 days (HEG104) (**A**) and 29 days (LEG110) (**D**). Vehicle or HCPT (4 or 8 mg/kg) was administered by paraneoplastic injection twice weekly (10 (for LEG104) or 8 (for LEG110) mice per group). The tumor volume was also measured twice weekly. *, *p* < 0.05; **, *p* < 0.01 indicates the significant decrease of tumor volume compared to the vehicle group. Data are shown as mean value ± SE. The tumor weight was recorded when the mice were sacrificed (**B**,**E**). The asterisks **, *p* < 0.01 indicates the significant decrease of body weight. The photographs show PDX tumors isolated from mice treated with HCPT (**C**,**F**). (**G**) IHC was performed for measuring the expression of Ki-67, TOP I, cleaved caspase-3, and H2A.XS139 in PDX tissue specimens; 5 to 7 independent tissue specimens were adopted for each group and analyzed (100× magnification). (**H**) Quantification results of responding IHC results. IOD values were measured by Image-pro Plus 6.0 software and shown as values ± SD. *, *p* < 0.05; **, *p* < 0.01; ***, *p* < 0.001 indicate a significant decrease in the indicated proteins in the treated tissues compared to the controls.

**Figure 6 cancers-11-01964-f006:**
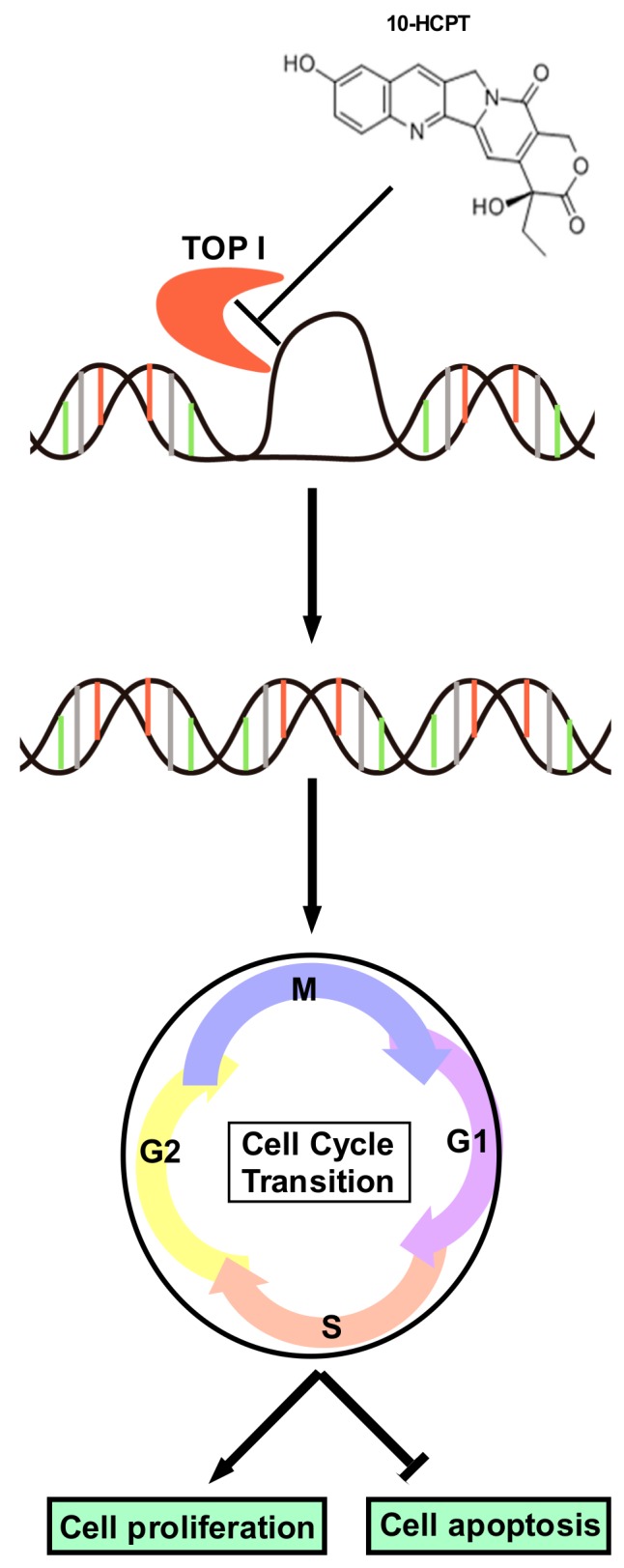
Schematic diagram of the mechanism of HCPT on ESCC. HCPT blocked the combination between TOP I enzyme and supercoiled DNA, decreasing the enzyme activity of TOP I. Foci formation subsequently occurred to promote DNA damage. As a consequence, cell cycle transition was inhibited and the G2/M phase was arrested. All of these processes finally led to cell proliferation halt and cell apoptosis, as indicated by the decreasing Ki-67 expression as well as the changing apoptosis markers.

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
