# Peer review of "(S)-10-Hydroxycamptothecin Inhibits Esophageal Squamous Cell Carcinoma Growth In Vitro and In Vivo Via Decreasing Topoisomerase I Enzyme Activity"

_cancers, 2019, doi:10.3390/cancers11121964_

Round 1

Reviewer 1 Report

Song et al. demonstrated the effect of 10-Hydroxycamptothecin (HCPT) on inhibiting esophageal squamous cell carcinoma growth.
Overall, the experimental design is good and the logic evolution is excellent. The authors hope to show the structure of HCPT in the figure 1. The approval number of the ethics committee must also be provided. In addition, information on the toxicity or pharmacokinetics of HCPT should be provided to describe the possibility of clinical application.

Reviewer 2 Report

Overall, this work does not provide novel mechanistic findings for the anti-cancer activity of HCPT.

Anticancer activity of (S)-10-Hydroxycamptothecin (HCPT), a TOP I specific inhibitor, has been reported in various cancers, which leads to the lack of scientific merits in this paper. While this paper provides in vitro and in vivo findings for the anti-esophageal cancer activity of HCPT, the authors fail to show any significant new mechanism for the anti-esophageal cancer activity different from its previous reported activities on other cancer types.

Reviewer 3 Report

The study finds that overexpression of topoisomerase (TOP) I in ESCC cells, and treatment with (S)-10-Hydroxycamptothecin (HCPT) inhibits the enzymatic activity TOP I at 24 h and decreases TOP I expression at 48 h and 72 h. HCPT also induces DNA damage by increasing the expression of H2A.XS139. HCPT significantly decreases the proliferation and anchorage-independent growth of esophageal squamous cell carcinoma (ESCC) cells (KYSE410, KYSE510, KYSE30 and KYSE450). Mechanistically, HCPT inhibits G2/M phase cell cycle transition, decreases the expression of cyclin B1 and elevates P21 expression. In addition, HCPT stimulates ESCC cells apoptosis, which is associated with elevated expression of cleaved PARP, cleaved caspase-3, cleaved caspase-7, Bax, Bim and inhibition of bcl-2 expression. HCPT dramatically inhibits patient-derived xenograft (PDX) tumor growth and decreases the expression of Ki-67 and TOP I, and increases the level of cleaved caspase-3 and H2A.XS139 expression. The followings need to be addressed. (1) In the figure 1, where do authors obtain/purchase tissue arrays? It should be described/detailed. (2) How many tumor tissues from patients do authors obtain to execute patient-derived xenograft (PDX) tumor model. And authors provide the IRB license when using clinical tissues from patients. (3) Authors should provide more data to prove that (S)-10-hydroxycamptothecin indeed inhibits esophageal squamous cell carcinoma growth via decreasing TOP I enzyme activity. Because (S)-10-hydroxycamptothecin might inhibit esophageal squamous cell carcinoma growth WITHOUT the relationship of TOP I activity. (4) Authors should use the TOP I specific inhibitor, siRNA or shRNA as positive control to confirm the induction of cell arrest and apoptosis due to the loss of TOP I activity. (5) Could TOP I overexpression in KYSE410, KYSE510, KYSE30 or KYSE450 cells overcome the effect of (S)-10-hydroxycamptothecin and increase cancer cell viability? (6) Authors should overexpress TOP I in normal SHEE cells to prove the important role of TOP I in in ESCC. (7) In the all figure legends, authors should provide the numbers of cases/experiments. (8) Dose (S)-10-hydroxycamptothecin (HCPT)-induce G2/M-phase arrest, and apoptosis…et al.(side effect) in normal SHEE cells? (9) Authors should provide the molecular weight of target proteins in western blotting. (10) In the figure 4B, authors should confirm the expression of Top I in KYSE410 cells with HCPT treatment for 48 hrs. It does not show the similar finding. (11) Authors should provide/describes the origin of SHEE cells? (12) Author could provide the cartoon figure presenting proposed mechanisms of (S)-10-hydroxycamptothecin (HCPT)-induced G2/M-phase arrest, and apoptosis in ESCC.

Reviewer 4 Report

Manuscript title: “(S)-10-Hydroxycamptothecin inhibits esophageal squamous cell carcinoma growth in vitro and in vivo via decreasing topoisomerase I enzyme activity

This is an interesting work investigating the use of (S)-10-Hydroxycamptothecin, a camptothecin derivative, as anticancer molecule in esophageal squamous cell carcinoma (ESCC). In particular, the authors evaluated for the first time the effects of topoisomerase (TOP) I inhibition both in vitro and in vivo. The results were interesting, showing that the decreased TOP I activity was related to the reduced cell proliferation and tumor growth, inhibition of G2/M transition and increased apoptosis.

The techniques used were appropriate and described with plenty details. Overall, this is a well-designed study with rigorous methods. The discussion is well-balanced, and the statements are supported by the data. The study is on a timely subject in view of increasing interest about the identification of new molecules with promise for use in anticancer therapy. However, there are some minor concerns to revise that are described below:

Introduction: Page 2 line 53: delete “incidence” (the incidence of ESCC does not depend on therapy) Page 2 lines 76-79: the final part should not anticipate any result. I suggest ending this section describing only the research objectives Results: Page 5 lines 157-159: unless strictly necessary, citations in Results section should be avoided. Therefore, I suggest to move this part in Discussion section. Discussion: The final part of Discussion section (page 9 lines 275-281) is a repetition of the Conclusion section (page 11 lines 383-388). Please delete one of the two sections.

To improve readability, some typos should be corrected (e.g. page 2 line 85 “cacner” and line 86 “Fgure”). Furthermore, I suggest some minor language corrections (e.g. page 1 use “chemoradiotherapy” instead of “chemoradiation therapy”; pages 8-9 use extended forms “cannot” and “did not” instead of contracted forms “can’t” and “didn’t”).

Author Response

Reviewer 4

This is an interesting work investigating the use of (S)-10-Hydroxycamptothecin, a camptothecin derivative, as anticancer molecule in esophageal squamous cell carcinoma (ESCC). In particular, the authors evaluated for the first time the effects of topoisomerase (TOP) I inhibition both in vitro and in vivo. The results were interesting, showing that the decreased TOP I activity was related to the reduced cell proliferation and tumor growth, inhibition of G2/M transition and increased apoptosis. The techniques used were appropriate and described with plenty details. Overall, this is a well-designed study with rigorous methods. The discussion is well-balanced, and the statements are supported by the data. The study is on a timely subject in view of increasing interest about the identification of new molecules with promise for use in anticancer therapy. However, there are some minor concerns to revise that are described below:

Comments:Introduction: Page 2 line 53: delete “incidence” (the incidence of ESCC does not depend on therapy) Page 2 lines 76-79: the final part should not anticipate any result. I suggest ending this section describing only the research objectives. Results: Page 5 lines 157-159: unless strictly necessary, citations in Results section should be avoided. Therefore, I suggest to move this part in Discussion section. Discussion: The final part of Discussion section (page 9 lines 275-281) is a repetition of the Conclusion section (page 11 lines 383-388). Please delete one of the two sections.To improve readability, some typos should be corrected (e.g. page 2 line 85 “cacner” and line 86 “Fgure”). Furthermore, I suggest some minor language corrections (e.g. page 1 use “chemoradiotherapy” instead of “chemoradiation therapy”; pages 8-9 use extended forms “cannot” and “did not” instead of contracted forms “can’t” and “didn’t”).

Response:We are very sorry about the mistakes. We have corrected the errors and carefully checked again as well as modified the manuscript according to your comments.

Round 2

Reviewer 3 Report

the authors have respond to my questions and revised the manuscript accordingly. I suggest it can be accepted for publication.

Author Response

Comment : The authors have respond to my questions and revised the manuscript accordingly. I suggest it can be accepted for publication.

Response : Thank you very much.